# COMPLEQA: Benchmarking the Impacts of Knowledge Graph Completion Methods on Question Answering

**Donghan Yu**
Carnegie Mellon University
dyu2@cs.cmu.edu

**Yu Gu**
The Ohio State University
gu.826@osu.edu

**Chenyan Xiong**
Carnegie Mellon University
cx@cs.cmu.edu

**Yiming Yang**
Carnegie Mellon University
yiming@cs.cmu.edu

## Abstract

How much success in Knowledge Graph Completion (KGC) would translate into the performance enhancement in downstream tasks is an important question that has not been studied in depth. In this paper, we introduce a novel benchmark, namely COMPLEQA, to comprehensively assess the influence of representative KGC methods on Knowledge Graph Question Answering (KGQA), one of the most important downstream applications. This benchmark includes a knowledge graph with 3 million triplets across 5 distinct domains, coupled with over 5,000 question-answering pairs and a completion dataset that is well-aligned with these questions. Our evaluation of four well-known KGC methods in combination with two state-of-the-art KGQA systems shows that effective KGC can significantly mitigate the impact of knowledge graph incompleteness on question-answering performance. Surprisingly, we also find that the best-performing KGC method(s) does not necessarily lead to the best QA results, underscoring the need to consider downstream applications when doing KGC[1].

## 1 Introduction

The inherent incompleteness of Knowledge Graphs (KGs) is a well-recognized problem. As such, a multitude of methods have been proposed to address this issue, spanning statistical relational learning (Getoor and Taskar, 2007), embedding-based models (Bordes et al., 2013; Yang et al., 2015; Sun et al., 2019b), neural-symbolic methods (Yang et al., 2017; Sadeghian et al., 2019; Qu et al., 2021), and recent language model-based methods (Yao et al., 2019; Wang et al., 2021; Saxena et al., 2022). However, prior studies have largely viewed KG completion as an end in itself, neglecting to investigate its potential impact on subsequent applications that utilize the completed KGs.

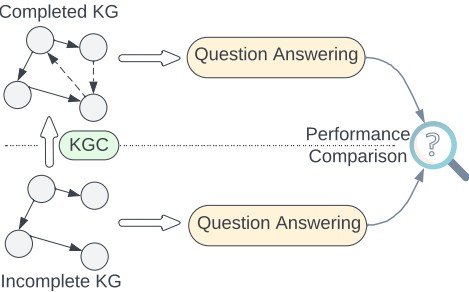

Figure 1: We compare the performance between incomplete and completed KGs by examining their question-answering results. Dashed arrows in the completed KG denote new triplets added by KGC.

The applications of KGs span different domains and tasks. Among them, Knowledge Graph Question Answering (KGQA), designed to answer natural language questions using information from KGs, is one of the most important. Extensive approaches have been proposed in recent years (Sun et al., 2019a; Verga et al., 2021; Ye et al., 2022; Gu and Su, 2022). Despite the rapid advancement, previous studies either assume the KGs are complete or handle incompleteness via specific question-answering methods (Saxena et al., 2020; Liu et al., 2022; Saxena et al., 2022), lacking a comprehensive investigation into different approaches. The impact of KGC on KGQA performance remains largely unexplored.

In our study, we seek to rectify this oversight by introducing a novel benchmark COMPLEQA designed to directly and holistically assess the influence of KG completion on KGQA. This benchmark comprises over three million triplets and approximately 400,000 entities across 5 different domains, collectively forming the KG. The corresponding QA dataset includes over 5,000 questions, featuring both single-hop and multi-hop questions. These questions are sorted into three generalization levels and are sourced from the GrailQA (Gu et al., 2021) dataset. For KG completion, we employ entity-centric incompleteness to align with the QA dataset,

---

[1]Our code and dataset will be released at https://github.com/PlusRoss/CompleQA

where the entities in the missing triplets correspond to those appearing in the questions. Importantly, different from previous studies, we actually incorporate predicted triplets into the KG, and use this completed KG for question answering, allowing a seamless study of various KGC and KGQA methods. Our investigation incorporates the study of four representative KG completion models, namely TransE (Bordes et al., 2013), DistMult (Yang et al., 2015), ComplEx (Trouillon et al., 2016), and RotatE (Sun et al., 2019b). For the KGQA models, we employ two published leading methods DecAF (Yu et al., 2023) and Pangu (Gu et al., 2022).

Our experiments show that the incompleteness of KGs can adversely affect QA performance, and effective KG completion can help alleviate this and improve performance by up to $6.1\%$ across varying degrees of incompleteness. Furthermore, we discern that KG completion performance does not always align with downstream performance, due to the limited overlap in triplet predictions between models and the disparate influence of these triplets on KGC and QA performance. We hope our findings stimulate more research that perceives KG completion not merely as an isolated objective but as a crucial step toward enhancing the performance of downstream tasks.

## 2 Benchmark Construction

In this section, we delineate the procedural steps employed in the development of our benchmark, including the knowledge graph, the question-answering dataset, and the completion dataset.

### 2.1 Knowledge Graph

A knowledge graph can be denoted as $\mathcal{KG} = (\{\mathcal{E}, \mathcal{R}, \mathcal{C}, \mathcal{L}\}, \mathcal{T})$, where $\mathcal{E}$ is the set of entities, $\mathcal{R}$ is the set of relations, $\mathcal{C}$ is the set of classes, $\mathcal{L}$ is the set of literals, and $\mathcal{T}$ is the set of triplets. A triplet, typically denoted as $(h, r, t)$, includes the head entity $h \in \mathcal{E}$, relation $r \in \mathcal{R}$ and tail entity or the respective class or literal $t \in (\mathcal{E} \cup \mathcal{C} \cup \mathcal{L})$.

The choice of the knowledge graph is a fundamental aspect, and in our study, we opt for Freebase (Bollacker et al., 2008), primarily owing to its widespread usage in academic circles. The original Freebase contains over $80$ million entities. To facilitate affordable exploratory research, we select a subset of Freebase as our final knowledge graph by specifically confining the KG to five important domains: *medicine*, *computer*, *food*, *law*,

Table 1: Data Statistics of the QA dataset.

| Question Category | Train | Valid | Test |
|---|---|---|---|
| Total | 3,395 | 997 | 973 |
| I.I.D. | - | 366 | 381 |
| Compositional | - | 232 | 225 |
| Zero-Shot | - | 399 | 367 |

Table 2: Data Statistics of the KG completion dataset. For the definition of Incompleteness, please refer to Section 2.3.

| Incompleteness | Train | Valid | Test |
|---|---|---|---|
| 20% | 1,674,405 | 1,807 | 2,864 |
| 50% | 1,667,397 | 4,519 | 7,160 |
| 80% | 1,660,390 | 7,230 | 11,456 |

and *business*. Within this sub-sample of Freebase, we encompass 395,965 entities, 383 relations, and 3,042,911 triplets. Despite the reduction in scale, we posit that this knowledge graph retains a substantial breadth and can accurately emulate real-world conditions and challenges. Following this, we explain the construction of the QA and completion datasets derived from this KG.

### 2.2 Question Answering over KG

We denote a QA model intended for $\mathcal{KG}$ as $f_{QA}(\mathcal{KG})$, and follow previous KGQA settings (Yih et al., 2016; Trouillon et al., 2016; Gu et al., 2021), where each natural language question $q$ aligns with an answer set $A$ and a logical form $l$, used to extract answers from the KG. We adopt GrailQA (Gu et al., 2021) as our foundation, which contains both single-hop and multi-hop questions and provides comprehensive evaluation over three generalization difficulty levels: i.i.d., compositional, and zero-shot, determined by whether the KG schemas that appeared in the test set have been observed in the training set. The original GrailQA is based on the full set of Freebase, thus we eliminate questions that cannot be answered or contradict the provided answer labels given our sub-sampled Freebase, This refinement process results in a total of $5,365$ questions with the statistical details in Table 1.

### 2.3 KG Completion

The completion task in a KG refers to, given an incomplete triplet $(h, r, ?)$ or $(?, r, t)$ where its head or tail entity is missing, the model is required to predict the missing entity[2].

To align it with the QA task for studying its impact, we first identify all the unique entities

---
[2]Following previous conventions, we don't consider the situation of missing classes or literals in this study.

from the validation and test questions[3], denoted as $E_{valid}$ and $E_{test}$ respectively. We then retrieve all triplets linked to these entities, denoted as $\mathcal{T}'_{valid}$ and $\mathcal{T}'_{test}$. We randomly choose a proportion $P$ of these triplets as the final validation and test sets, $\mathcal{T}_{valid}$ and $\mathcal{T}_{test}$ for KG completion. All remaining triplets, including those unlinked to those entities, form the training data $\mathcal{T}_{train}$. We adjust $P$ to be 20%, 50%, and 80% to introduce varying degrees of incompleteness. The quantity of triplets is detailed in Table 2. Note that compared with random sampling from the entire KG, sampling triplets based on entities appeared in the questions can better align the two tasks. Furthermore, this is also seen as a realistic scenario where relatively new entities in KGs often come with incomplete triplets and their related questions are both important and challenging.

## 2.4 Effect of KG Completion over KGQA

To study the impact of knowledge graph completion on question answering, we incorporate the completed triplet into the KG, then compare the QA performance using the original incomplete KG with the completed KG. This allows us to freely explore any KGC and question-answering methods. The process of incorporating triplets is detailed below.

In the case of each incomplete triplet $(h, r, ?)$, the model first predicts $N$ candidate tail entities $t_i$ with scores $s_i$ denoted as $[(t_1, s_1), \cdots, (t_N, s_N)]$, where $s_1 \geq \cdots \geq s_N$. To determine whether triplet $(h, r, t_i)$ should be incorporated into the incomplete KG, we establish a threshold $s_T$. If the score $s_i \geq s_T$, the triplet $(h, r, t_i)$ is added to the KG. Note that we don't add the triplet which is already in the KG. The same process is followed for missing head entity $(?, r, t)$. Suppose $\mathcal{T}^{pred}_{valid}$ represents the collection of all added triplets for the validation set and $\mathcal{T}^{pred}_{test}$ for the test set. $\mathcal{KG}(\mathcal{T})$ represents the KG that maintains the consistent set of entities, relations, classes, and literals but incorporates a variable set of triplets $\mathcal{T}$. Finally, we evaluate the performance of the incomplete KG $f_{QA}(\mathcal{KG}(\mathcal{T}_{train}))$ versus the completed KG $f_{QA}(\mathcal{KG}(\mathcal{T}_{train} \cup \mathcal{T}^{pred}_{valid} \cup \mathcal{T}^{pred}_{test}))$. The performance difference can indicate the utility of KG completion over the QA task. Further details on the utilized models, evaluation metrics, and how to determine $s_T$ are provided in the following section.

---

[3]GrailQA provides annotated labels for those entities.

## 3 Methods and Evaluation Metrics

This section presents the methods implemented on our benchmark and the evaluation metrics employed to assess their effectiveness.

## 3.1 Question Answering

We employ two state-of-the-art methods, namely DecAF (Yu et al., 2023) and Pangu (Gu et al., 2022). For simplicity, we employ oracle entity linking to Pangu, avoiding the need to train a specially designed entity linker. In contrast, DecAF operates based on text retrieval without the need for entity linking. Due to computational limitations, we chose T5-base (Raffel et al., 2020) as the backbone model for DecAF and BERT-base (Devlin et al., 2019) for Pangu. F1 scores of answer matching are used for the evaluation of QA performance.

## 3.2 KG Completion

The KGC methods employed in this study comprise four representative models: TransE (Bordes et al., 2013), DistMult (Yang et al., 2015), ComplEx (Trouillon et al., 2016), and RotatE (Sun et al., 2019b). For the implementation of these methods, we turn to LibKGE (Broscheit et al., 2020), a highly developed KGC library, with details introduced in Appendix A.

To measure the performance of KGC, we adhere to standard metrics for missing entity prediction: Mean Reciprocal Rank (MRR) and Hits@K (H@K) in the filtered setting (Bordes et al., 2013). Importantly, considering our goal to incorporate predicted triplets into the KG for subsequent question answering, we propose to measure triplet prediction by F1 score: F1 $= 2 \cdot \frac{|\mathcal{T}_{pred} \cap \mathcal{T}_{gold}|}{|\mathcal{T}_{pred}| + |\mathcal{T}_{gold}|}$ where $\mathcal{T}_{pred}$ represents the set of predicted triplets while $\mathcal{T}_{gold}$ denotes the set of missing ground-truth triplets. We adjust the threshold $s_T$ introduced in Section 2.4 on the validation split to achieve the best F1 score. We show such adjustment also produces good QA performance in Appendix B.

## 4 Experiments

In this section, we delve into the outcomes of our empirical investigations. Firstly, from Table 3, it's clear that the KG incompleteness negatively affects performance. Specifically, the F1 scores of DecAF and Pangu drop by 14.0% and 12.1% respectively when the KG incompleteness level is 50%. Subsequently, we aim to address the following questions:

Table 3: Experiment results on KG completion and question answering of our benchmark COMPLEQA. Results are averaged across 3 independent runs with different random seeds. "QA w/ X" signifies the QA performance using model X. The top performance in each column of each section is highlighted in **bold**. For QA performance, we provide a percentage change against scenarios where no KG completion was used, which is color-coded: green for positive changes and red for negative ones.

| Incompleteness | Model | KGC | | | | | QA w/ DecAF | QA w/ Pangu |
| --- | --- | --- | --- | --- | --- | --- | --- | --- |
| | | MRR | H@1 | H@3 | H@10 | F1 | F1 | F1 |
| 0% | - | - | - | - | - | - | 0.782 | 0.880 |
| 20% | None | - | - | - | - | - | 0.742 | 0.846 |
| | TransE | 0.614 | 0.533 | 0.667 | 0.761 | 0.467 | 0.750 ↑1.1% | 0.864 ↑2.1% |
| | RotatE | 0.667 | 0.612 | 0.700 | **0.763** | 0.573 | **0.765** ↑3.1% | 0.864 ↑2.1% |
| | DistMult | 0.671 | 0.618 | 0.718 | 0.754 | 0.637 | 0.757 ↑2.0% | 0.865 ↑2.2% |
| | ComplEx | **0.681** | **0.630** | **0.722** | 0.756 | **0.660** | 0.757 ↑2.0% | **0.868** ↑2.6% |
| 50% | None | - | - | - | - | - | 0.672 | 0.773 |
| | TransE | 0.462 | 0.392 | 0.500 | 0.590 | 0.439 | 0.684 ↑1.8% | 0.765 ↓1.0% |
| | RotatE | **0.509** | 0.455 | **0.538** | **0.607** | 0.524 | 0.701 ↑4.3% | 0.778 ↑0.6% |
| | DistMult | 0.506 | **0.471** | 0.528 | 0.564 | 0.565 | **0.713** ↑6.1% | **0.798** ↑3.2% |
| | ComplEx | 0.504 | 0.467 | 0.527 | 0.564 | **0.582** | 0.712 ↑6.0% | **0.798** ↑3.2% |
| 80% | None | - | - | - | - | - | 0.586 | 0.638 |
| | TransE | 0.299 | 0.240 | 0.324 | **0.408** | 0.335 | 0.578 ↓1.4% | **0.650** ↑1.9% |
| | RotatE | **0.312** | **0.265** | **0.328** | 0.403 | **0.373** | **0.591** ↑0.9% | 0.636 ↓0.3% |
| | DistMult | 0.255 | 0.218 | 0.270 | 0.320 | 0.311 | 0.585 ↓0.2% | 0.640 ↑0.3% |
| | ComplEx | 0.252 | 0.212 | 0.272 | 0.323 | 0.320 | 0.588 ↑0.3% | **0.650** ↑1.9% |

**Q1: How much does the good performance in KGC translate to the enhancement in QA?** The experimental results in Table 3 demonstrate that KGC can indeed enhance the QA performance in the majority of the cases. Notably, the ComplEx algorithm for KGC boosted the QA performance of DecAF alone (without KGC) by 6.0% and of Pangu by 3.2% at 50% incompleteness level. However, at the 80% incompleteness level, the same ComplEx algorithm only boosted the QA performance of DecAF by 0.3% and of Pangu by 1.9%, which are much less than the performance gains at the 50% and 20% incompleteness levels. The observations on other KGC methods demonstrated similar patterns; some of them even lead to decreased QA results. This could be attributed to that incorrect triplets introduced by the KGC methods outweigh the correctly predicted ones.

To validate this, we construct a scenario that incorporates only the correctly-predicted triplets into the KG, while all the incorrect triplets are discarded. Figure 2, along with other results in Appendix C, clearly illustrates a significant performance enhancement in this scenario, especially at the 80% incompleteness level, thereby substantiating the detrimental impact of incorrect predicted triplets on the QA model.

**Q2: Does the best KGC method(s) always lead to the best QA performance?** According to Table 3, we see that better KGC does not always translate to better downstream outcomes. As an illustration,

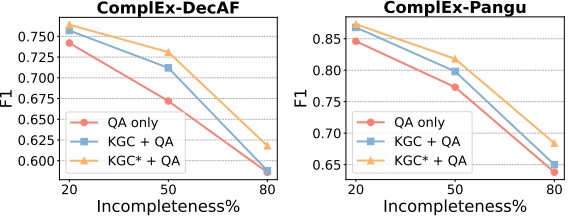

Figure 2: QA performance under different completed KGs. "QA only" means using the original incomplete KG for QA. "KGC + QA" means using the KGC-completed KG. KGC* means only keeping the correct triplets while discarding all the incorrect ones.

although ComplEx achieves exceptional KGC results at 20% incompleteness, it does not outperform RotatE in terms of QA performance when DecAF is used. Similarly, even though RotatE produces top-tier KGC outcomes at 80% incompleteness, it even leads to the worst QA performance when Pangu is utilized.

To quantitatively assess the relationship between the performance of KGC and QA, we use Spearman's rank correlation coefficient. This measures the Pearson correlation between the ranked values of two variables. In many instances, we find that the correlation between the KGC metric and QA metric is not high, with a value below 0.5. Another observation is that the F1 score of KGC performance corresponds better with QA performance than the mean reciprocal rank (MRR). This is reasonable because QA performance relies on

Table 4: Spearman's rank correlation coefficient between KGC metrics (MRR and F1) and QA metrics (F1) with two models DecAF and Pangu.

| Incompleteness | Metrics | F1-DecAF | F1-Pangu |
|---|---|---|---|
| 20% | MRR | 0.32 | 0.95 |
|  | F1 | 0.32 | 0.95 |
| 50% | MRR | 0.40 | 0.32 |
|  | F1 | 0.80 | 0.95 |
| 80% | MRR | 0.20 | -0.63 |
|  | F1 | 0.40 | -0.30 |

the completed knowledge graph, which includes predicted triplets. The F1 score, in this context, is a more direct measure of the quality of predicted triplets compared to MRR.

To delve further, we measured the overlap of predicted triplets among various models and found that no model's predictions, correct or incorrect, fully encompassed the other's. For example, in a 20% incomplete KG, ComplEx and RotatE had about 80% shared correct predictions and 30% shared incorrect ones. In this case, despite ComplEx's superior performance in KGC, it doesn't solely determine QA performance as various predicted triplets impact QA differently, and this impact may not align well with their contribution to KGC performance. This discrepancy points to the need for KGC methods that optimize both KG completion and downstream task performance.

## 5 Conclusion

In this study, we introduced a novel benchmark to investigate the impact of representative KGC methods on the Knowledge Graph Question Answering (KGQA) task. Our findings demonstrate that KG incompleteness negatively affects KGQA, and effective KGC can significantly mitigate this issue. However, we also discovered that best-performing KGC method does not necessarily lead to the best KGQA results. Our work underlines the necessity to view KGC not merely as a standalone goal, but as a vital step toward improving downstream tasks.

## Limitations

While our study offers new insights into the relationship between Knowledge Graph Completion (KGC) and Knowledge Graph Question Answering (KGQA), it has several limitations that require further exploration.

The primary focus of our study was to examine the influence of KGC on KGQA, which, although an essential application of KGs, is just one of many

potential downstream tasks. Other tasks like recommendation systems and semantic search may also benefit from KGC, and their responses to KGC could differ from KGQA. Therefore, more research is needed to fully comprehend the impact of KGC on these diverse applications.

Additionally, our study did not employ large language models for few-shot learning in question answering, a technique that is in vogue in the field of NLP. Assessing the impact of KGC in such scenarios would provide fascinating insights and help determine if our conclusions hold true in these trending methods.

## Ethics Statement

An important ethical concern arising from our research is the potential misuse of Knowledge Graphs (KGs) augmented by Knowledge Graph Completion (KGC) methods. While KGC aims to enhance KG completeness and boost Question Answering (QA) systems' accuracy, it may inadvertently introduce or propagate biases or inaccuracies if the KGC algorithms exhibit inherent flaws.

Such biases could affect QA systems, leading to skewed or misleading responses that can foster misinformation or reinforce existing biases, especially in critical decision-making domains. This underlines the need for responsible use and continuous evaluation of KGC algorithms, involving rigorous validation of predicted triplets and transparency about algorithmic workings and limitations. Future work must focus on devising methods to identify, measure, and rectify biases within KGC algorithms, promoting ethical use of these technologies.

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

# A Knowledge Graph Completion

We introduce the hyper-parameter settings for our KG completion models based on the LibKGE (Broscheit et al., 2020) library.

The dimensionality of our embeddings is set to 200, utilizing xavier uniform initialization (Glorot and Bengio, 2010). Following previous approaches (Bordes et al., 2013; Sun et al., 2019b), we employ negative sampling for training. In this process, negative triplets are created for every triplet in the training dataset by randomly substituting either the head or tail entity. Each triplet gives rise to 50 negative triplets, half by replacing the head entity and half by substituting the tail. The batch size is 4096, with a maximum of 500 epochs. We deploy early stopping after 5 evaluations over the validation set, with one evaluation after every 10 epochs.

When it comes to optimization, we leverage Adam (Kingma and Ba, 2015) for TransE (Bordes et al., 2013) and RotatE (Sun et al., 2019b) with a learning rate of 0.001. For ComplEx (Trouillon et al., 2016) and DistMult (Yang et al., 2015), we employ Adagrad (Duchi et al., 2010) with a learning rate within the range [0.01, 1.0].

For RotatE and TransE, we choose from binary cross entropy (BCE), Kullback-Leibler divergence (KL), and margin ranking (MR) for the training losses, and explore the loss argument within [0, 50]. For ComplEx and DistMult, we utilize BCE loss with a loss argument of 0.0. We experimented with other training losses and arguments, but found them ineffective. Hyperparameter search is conducted via Bayesian optimization using the Ax framework (https://ax.dev/), with the number of trials as 40.

# B Score Threshold

We aimed to study the effect of score threshold $s_T$ for each KGC method, which is introduced in Section 2.4. We vary this threshold for each method to evaluate the corresponding KGC and QA performance. As shown in Figure 3, the KGC F1 score initially rises and then falls, as increasing the score threshold leads to fewer triplets being added, which although improves precision, it negatively impacts recall. We see that the curve representing the relationship between the score threshold and the KGC F1 score appears to coincide well with the QA performance curve. However, there are instances where this alignment is less pronounced, indicating that utilizing KGC F1 to pinpoint the task-specific score threshold provides a useful starting point, but

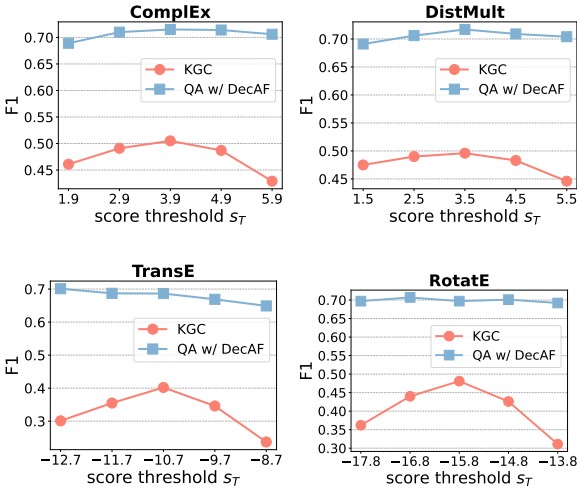

Figure 3: Performance variation with different score threshold. Each plot depicts the performance trends for one KGC method at $50\%$ KG incompleteness level. The curves represent changes in KGC F1 scores on the validation split and QA w/ DecAF F1 scores on the test split as the score threshold varies.

is not sufficient on its own. We believe that further investigation of this issue could provide valuable insights for future research.

## C Filtering Incorrect Triplets

As additional experiments for Section 4.1, we present the question-answering performance of other KGC methods besides ComplEx after discarding incorrect triplets and retaining only the correct ones. As depicted in Figure 4, it's clear that QA efficacy significantly improves when incorrect triplets are excluded, notably when the knowledge graph is $80\%$ incomplete, which reflects a situation where KGC quality is notably poor, with a high frequency of incorrect triplet predictions.

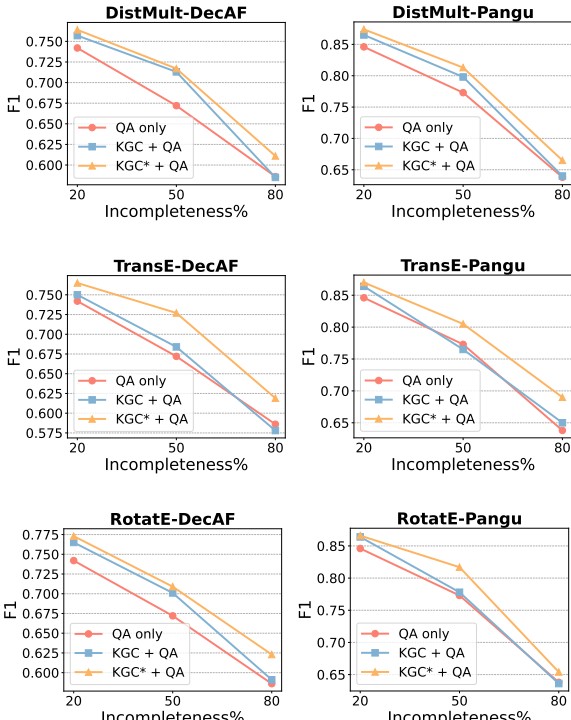

Figure 4: QA performance under different completed KGs. "QA only" means using the original incomplete KG for QA. "KGC + QA" means using the KGC-completed KG. KGC* means only keeping the correct triplets while discarding all the incorrect ones.