# OpenReview forum: "CompleQA: Benchmarking the Impacts of Knowledge Graph Completion Methods on Question Answering"
_EMNLP/2023/Conference — EMNLP 2023 Findings_

### Official Review · Reviewer_81Bu · 2023-08-03

**Soundness:** 2

**Excitement:**

2: Mediocre: This paper makes marginal contributions (vs non-contemporaneous work), so I would rather not see it in the conference.

**Paper Topic And Main Contributions:**

This paper investigates how the incompletion of KG influence downstream QA task. They found incompletion usually leads to worse QA performance but sometimes improves it.

**Questions For The Authors:**

The correlation Between KGC and QA is a good result. But how do you propose to solve this rather than pure observation?

**Reasons To Accept:**

The experimental results seem solid.
The writing is clear and easy to follow
The question is interesting and worth investigation

**Reasons To Reject:**

1. The overall setting is weird to me. The incompletion ratio is set to 20, 50, 80%. however, how would one opt to such a KG system with such an extremely high incompletion ratio? This is not realistic.
2. The incompletion in KG can sometimes lead to improvement in QA. This is an interesting finding, but how would you support the QA system with this finding? I did not see a deeper analysis of leveraging this and how to overcome the drawbacks.

**Reproducibility:**

4: Could mostly reproduce the results, but there may be some variation because of sample variance or minor variations in their interpretation of the protocol or method.

**Reviewer Confidence:**

4: Quite sure. I tried to check the important points carefully. It's unlikely, though conceivable, that I missed something that should affect my ratings.

---

### Official Review · Reviewer_TkCf · 2023-08-04

**Soundness:** 4

**Excitement:**

4: Strong: This paper deepens the understanding of some phenomenon or lowers the barriers to an existing research direction.

**Paper Topic And Main Contributions:**

This paper proposes a benchmark that investigates the influence of knowledge graph completion (KGC) on knowledge graph question answering (KGQA), as a prominent application. The paper builds on existing datasets to create a dataset specifically for this benchmark. The experiments test the robustness of use state-of-the-art KGC and their influence on KGQA methods. The paper suggests that effective KGC can improve KGQA but often better KGC performance does not align with better KGQA performance.

**Reasons To Accept:**

* The paper is well written and motivated, and it is easy to follow.
* The experimental design is sound and detailed.
* Interesting and novel task that has been overlooked in the literature. Has potential on further defining the practical usefulness of KGC by influencing the way of evaluating of KGC in the context of practical applications.

**Reasons To Reject:**

The paper lacks analysis that would give insights on why better KGC performance does not align with better KGQA performance. This includes quantitative analysis, e.g,. the entity types / relations for which this phenomenon is more frequent, and qualitative analysis, e.g., questions that are anyway harder to answer even if the correct triples have been predicted by KGC. Since this is a short paper, this could be included in the Appendix.

**Reproducibility:**

4: Could mostly reproduce the results, but there may be some variation because of sample variance or minor variations in their interpretation of the protocol or method.

**Reviewer Confidence:**

4: Quite sure. I tried to check the important points carefully. It's unlikely, though conceivable, that I missed something that should affect my ratings.

---

### Official Review · Reviewer_kWfn · 2023-08-10

**Typos Grammar Style And Presentation Improvements:** 048
**Soundness:** 3

**Excitement:**

3: Ambivalent: It has merits (e.g., it reports state-of-the-art results, the idea is nice), but there are key weaknesses (e.g., it describes incremental work), and it can significantly benefit from another round of revision. However, I won't object to accepting it if my co-reviewers champion it.

**Paper Topic And Main Contributions:**

Paper is about KG completion (KGC) and impact on downstream question answering. It aims to motivate the KGC problem by showing that KGC impacts QA performance, which is a significant application of KGs. Contributions are NLP engineering experiment and data resource derived from existing artifacts (GrailQA, Freebase). Specifically:
1) Argumentation for the important of grounding the KGC task in its impact on downstream performance
2) Benchmark for measuring KGC task benefit on KGQA using a small subset of Freebase (400K entities out of 80M) and questions mined from GrailQA
3) Empirical comparison of popular KGC techniques on 2 KGQA techniques (Pangu, DecAF).



**Questions For The Authors:**

1) Can you give more intuition behind the choice of Pangu and DecAF as proposed KGQA baselines? Is there a reason that a semantic parsing technique was not considered? I accept the LLM explanation in Limitations but am skeptical at the lack of any representation of semantic parsing techniques for this task.

2) Did you consider weighting obscured triples by predicate or some other strategy to control the rate that KGC models get credit for completing irrelevant triples?

3) Is it possible to include examples of incomplete triples that were filled and subsequently resulted in a correct output? I.e. do you have/can you include some examples where KGQA is incorrect because of incompleteness and then made correct by KGC?

**Reasons To Accept:**

A strength of this paper is its focus on grounding a long studied academic Graph task (KGC) in a long standing downstream NLP task (KGQA). The intuition to ground a task in its benefits to other tasks gives weight to the importance of their proposal on application in this domain. Additionally, the authors start from a widely cited and popular benchmark (GrailQA) that includes questions from IID, compositional, and zero-shot domains; this adds empirical weight to their findings because this is widely thought to be a high quality benchmark. Lastly, the authors select a representative set of popular KGC methods. In terms of writing, this paper is well written and easy to follow.

**Reasons To Reject:**

I think there are two reasons to consider rejecting this paper:
1) The notion of incompleteness for their KGC task is somewhat artificial; the authors obtain relations of answer entities in dev/test set, and  "incomplete" them by obscuring one of the entities in the relation, and then "randomly choose a proportion P of these triplets as the final validation and test sets". This challenge here is that not all triples of answer entities are equally meaningful for QA; i.e. it is possible that some and probably likely that some, if not most, triples, are largely meaningless. As a result, obscuring them would be expected to have little/no impact on QA. This risks significantly diluting the correlation between KGC quality and downstream QA. Specifically, a KGC model might look great because it fills in lots of relations but then look bad for QA because some large percent of those relations were meaningless. This can potentially mislead the reader into thinking that a particular KGC model is strong on KGC but bad for QA, when in fact the measurement of KGC model strength on this benchmark may just be an artifact of the benchmark. I think this can be improved by applying stratified sampling over predicates to ensure that we are controlling not just for the entities we obscure but also for relative importance of the information the KGC model completes. Further motivating the need for a deeper look into this approach is Appendix D, which seems to indicate a weak correlation between KGC and KGQA. It would be interesting to see the set of questions for which there is little to no correlation, as this is where the most opportunity is to improve this benchmark. Ultimately I view this as something that dilutes, but does not invalidate, the conclusions. There is a correlation, and that correlation needs to be explored further, but unfortunately the authors don't dig into the link between KGC and KGQA much to explain this correlation.

2) Modeling techniques studied; the authors study two KGQA techniques, Pangu and DecAF. While both exhibit strong performance, these are not the most natural nor most representative baseline choices. Neither of these techniques are popular or widely cited; DecAF is somewhat popular and adopts an IR approach to KGQA, in which documents of triples are retrieved as evidence to inform answer generation, while Pangu is not and employs a neurosymbolic planner/critic pipeline. This paper would be made more sound by the inclusion of more popular and representative techniques, especially those of the semantic parsing variety, of which there are many popular approaches (citations below). The lack of semantic parsing baseline is concerning for a few reasons, including that 1) these are the most widely studied approach to KGQA and 2) semantic parsing provides a more direct means to assess KGC. W/r/t 2) specifically, these approaches would enable the authors to measure the benefit of KGC on generated parse success rate. For example, using a parsing approach trained on the complete KG and evaluated on the incomplete one would provide a realistic measurement of the impact of missing data, as parses that seek to return the obscured data would no longer be executable. The authors should give more justification behind why they chose DecAF and Pangu, beyond just "here are 2 SOTA techniques". While they discuss the lack of an LLM-only baseline in the limitation, they don't include semantic parsing.

@article{Ye2021RNGKBQAGA,
  title={RNG-KBQA: Generation Augmented Iterative Ranking for Knowledge Base Question Answering},
  author={Xi Ye and Semih Yavuz and Kazuma Hashimoto and Yingbo Zhou and Caiming Xiong},
  journal={ArXiv},
  year={2021},
  volume={abs/2109.08678},
  url={https://api.semanticscholar.org/CorpusID:237562927}
}

@inproceedings{Abdelaziz2021ASP,
  title={A Semantic Parsing and Reasoning-Based Approach to Knowledge Base Question Answering},
  author={I. Abdelaziz and Srinivas Ravishankar and Pavan Kapanipathi and Salim Roukos and Alexander G. Gray},
  booktitle={AAAI Conference on Artificial Intelligence},
  year={2021},
  url={https://api.semanticscholar.org/CorpusID:235363625}
}

**Reproducibility:**

4: Could mostly reproduce the results, but there may be some variation because of sample variance or minor variations in their interpretation of the protocol or method.

**Reviewer Confidence:**

4: Quite sure. I tried to check the important points carefully. It's unlikely, though conceivable, that I missed something that should affect my ratings.

---

### Official Review · Reviewer_dwAq · 2023-08-11

**Soundness:** 3

**Excitement:**

2: Mediocre: This paper makes marginal contributions (vs non-contemporaneous work), so I would rather not see it in the conference.

**Paper Topic And Main Contributions:**

The paper presents a benchmark/benchmarking approach to evalaute the impact of predictions made by knowledge graph embeddings on question answering. The benchmark datasets consists of 3M triples with 5K questions. The introduction references a large number of KGQA approaches and corresponding benchmarking. The authors then delve into the construction of their benchmark. In particular, the KG completion is carried out using embedding methods. The evaluation with two state-of-the-art approaches suggests that increasing the incompleteness leads to worse results. Adding only correct triples tends to improve the results but in some cases, adding predicted triples actually worsens the performance.

**Reasons To Accept:**

- Interesting research area
- Combination of results from different domains (QA, embedddings)


**Reasons To Reject:**

- No major scientific contribution
- No surprising insights
- Results achieved on one dataset

**Reproducibility:**

4: Could mostly reproduce the results, but there may be some variation because of sample variance or minor variations in their interpretation of the protocol or method.

**Reviewer Confidence:**

4: Quite sure. I tried to check the important points carefully. It's unlikely, though conceivable, that I missed something that should affect my ratings.

---

### Meta-Review · Area_Chair_ZAg5 · 2023-09-19

**Recommendation:** 3

**Metareview:**

**Summary:**
The paper introduces a benchmarking approach to assess the impact of predictions generated by knowledge graph embeddings on question answering. The benchmark dataset comprises 3 million triples and 5,000 questions. The experiments aim to evaluate the robustness of state-of-the-art knowledge graph completion (KGC) methods and their influence on knowledge graph question answering (KGQA) techniques.
The authors provide insights into the construction of their benchmark, emphasizing the use of embedding methods for knowledge graph completion. The goal is to highlight the relevance of the KGC problem by demonstrating its impact on question answering performance, which is a significant application of knowledge graphs. The contributions of this work include NLP engineering experiments and a data resource derived from existing artifacts such as GrailQA and Freebase.
The evaluation results suggest that increasing the incompleteness of the knowledge graph leads to worse performance. While the addition of correct triples tends to improve results, there are cases where the inclusion of predicted triples actually worsens performance.

**Strengths:**
A notable strength of this paper lies in its approach to bridging a well-established academic task, knowledge graph completion, with a longstanding practical NLP task, namely, the impact of knowledge graph embeddings on question answering.
The authors' insight into connecting a task with its potential benefits for other tasks underscores the significance of their proposal within this domain.
Furthermore, the authors build upon a highly regarded benchmark, GrailQA, which encompasses questions from various domains, including IID, compositional, and zero-shot, lending empirical robustness to their findings, as GrailQA is widely recognized as a high-quality benchmark. Additionally, the authors carefully select a representative set of popular knowledge graph completion (KGC) methods and techniques, including Pangu and DecAF.

**Weaknesses:**
Reviewers have identified the following weaknesses in this work:
1. The concept of "incompleteness" in the knowledge graph completion (KGC) task appears somewhat artificial. The authors create this sense of incompleteness by obtaining relations of answer entities in the dev/test set and then obscuring one of the entities in the relation. Subsequently, they randomly select a proportion (P) of these triplets as the final validation and test sets. This approach might inadvertently lead readers to conclude that a specific KGC model has good performance in KGC but performs poorly in QA, when in reality, the observed KGC model strength on this benchmark may be an artifact of the benchmark construction.
2. While there is a correlation between KGC and knowledge graph question answering (KGQA), this correlation remains largely unexplored. The paper lacks a thorough analysis that would provide insights into why superior KGC performance does not necessarily translate to better KGQA performance. This analysis could encompass quantitative factors, such as entity types and relations more prone to this phenomenon, as well as qualitative aspects, like questions that remain challenging to answer even if the correct triples are predicted by KGC.
3. The paper could benefit from the inclusion of more popular and representative techniques, particularly those related to semantic parsing. The absence of a semantic parsing baseline is a concern for several reasons, including the widespread use of semantic parsing in KGQA and its suitability for assessing KGC. Incorporating such approaches would enable the authors to measure the impact of KGC on generated parse success rates.
4. The selection of DecAF and Pangu as techniques for evaluation should be substantiated further. The authors should provide more justification beyond simply stating that these are two state-of-the-art techniques. Additionally, while the limitation section mentions the absence of an LLM-only baseline, the paper does not incorporate semantic parsing, which is a notable omission.
5. The overall experimental setting, particularly the choice of incompletion ratios at 20%, 50%, and 80%, might not align with a realistic KG system with such an extremely high incompletion ratio. Further context or justification for these ratios would enhance the paper's credibility.

**Author-Reviewer discussion and acknowledgment:**
The authors have not provided any rebuttal comments.

**Conclusion:**
This paper is well-written and easy to follow. The contribution is interesting, addressing a task that has been overlooked in the literature. However, reviewers recommend that the authors incorporate more state-of-the-art research and consider additional improvements. It is suggested that the work should include additional data based on the points highlighted within the reviews. Additionally, the authors should address the identified typos in the paper.

---

### Decision · Program_Chairs · 2023-10-07

**Decision:**

Accept-Findings

**Comment:**

**Summary:**
The paper introduces a benchmarking approach to assess the impact of predictions generated by knowledge graph embeddings on question answering. The benchmark dataset comprises 3 million triples and 5,000 questions. The experiments aim to evaluate the robustness of state-of-the-art knowledge graph completion (KGC) methods and their influence on knowledge graph question answering (KGQA) techniques.
The authors provide insights into the construction of their benchmark, emphasizing the use of embedding methods for knowledge graph completion. The goal is to highlight the relevance of the KGC problem by demonstrating its impact on question answering performance, which is a significant application of knowledge graphs. The contributions of this work include NLP engineering experiments and a data resource derived from existing artifacts such as GrailQA and Freebase.
The evaluation results suggest that increasing the incompleteness of the knowledge graph leads to worse performance. While the addition of correct triples tends to improve results, there are cases where the inclusion of predicted triples actually worsens performance.

**Strengths:**
A notable strength of this paper lies in its approach to bridging a well-established academic task, knowledge graph completion, with a longstanding practical NLP task, namely, the impact of knowledge graph embeddings on question answering.
The authors' insight into connecting a task with its potential benefits for other tasks underscores the significance of their proposal within this domain.
Furthermore, the authors build upon a highly regarded benchmark, GrailQA, which encompasses questions from various domains, including IID, compositional, and zero-shot, lending empirical robustness to their findings, as GrailQA is widely recognized as a high-quality benchmark. Additionally, the authors carefully select a representative set of popular knowledge graph completion (KGC) methods and techniques, including Pangu and DecAF.

**Weaknesses:**
Reviewers have identified the following weaknesses in this work:
1. The concept of "incompleteness" in the knowledge graph completion (KGC) task appears somewhat artificial. The authors create this sense of incompleteness by obtaining relations of answer entities in the dev/test set and then obscuring one of the entities in the relation. Subsequently, they randomly select a proportion (P) of these triplets as the final validation and test sets. This approach might inadvertently lead readers to conclude that a specific KGC model has good performance in KGC but performs poorly in QA, when in reality, the observed KGC model strength on this benchmark may be an artifact of the benchmark construction.
2. While there is a correlation between KGC and knowledge graph question answering (KGQA), this correlation remains largely unexplored. The paper lacks a thorough analysis that would provide insights into why superior KGC performance does not necessarily translate to better KGQA performance. This analysis could encompass quantitative factors, such as entity types and relations more prone to this phenomenon, as well as qualitative aspects, like questions that remain challenging to answer even if the correct triples are predicted by KGC.
3. The paper could benefit from the inclusion of more popular and representative techniques, particularly those related to semantic parsing. The absence of a semantic parsing baseline is a concern for several reasons, including the widespread use of semantic parsing in KGQA and its suitability for assessing KGC. Incorporating such approaches would enable the authors to measure the impact of KGC on generated parse success rates.
4. The selection of DecAF and Pangu as techniques for evaluation should be substantiated further. The authors should provide more justification beyond simply stating that these are two state-of-the-art techniques. Additionally, while the limitation section mentions the absence of an LLM-only baseline, the paper does not incorporate semantic parsing, which is a notable omission.
5. The overall experimental setting, particularly the choice of incompletion ratios at 20%, 50%, and 80%, might not align with a realistic KG system with such an extremely high incompletion ratio. Further context or justification for these ratios would enhance the paper's credibility.

**Author-Reviewer discussion and acknowledgment:**
The authors have not provided any rebuttal comments.

**Conclusion:**
This paper is well-written and easy to follow. The contribution is interesting, addressing a task that has been overlooked in the literature. However, reviewers recommend that the authors incorporate more state-of-the-art research and consider additional improvements. It is suggested that the work should include additional data based on the points highlighted within the reviews. Additionally, the authors should address the identified typos in the paper.